# Phase angle is related to oxidative stress and antioxidant biomarkers in breast cancer patients undergoing chemotherapy

**Bruna R. da Silva**[1]*, **Sarah Rufato**[1], **Mirele S. Mialich**[1], **Loris P. Cruz**[2], **Thais Gozzo**[2], **Alceu A. Jordão**[1]

1 Department of Health Sciences, Ribeirão Preto Medical School, University of São Paulo (USP), Ribeirão Preto, São Paulo, Brazil, 2 Nursing School of Ribeirão Preto, University of São Paulo, Ribeirão Preto, São Paulo, Brazil

* bruna.ramos.silva@alumni.usp.br

**Data Availability Statement:** All relevant data are within the paper.

**Funding:** RS was founded by São Paulo Research Foundation (FAPESP). Scholarship number: 2017/

## Abstract

### Purpose

The study aimed to analyze the influence of chemotherapy on health biomarkers and examine the relationship between phase angle (PhA) and oxidative stress.

### Methods

A prospective study was performed. Women who were starting chemotherapy were recruited. Also, this study included a control group of women without cancer. Bioelectrical impedance multiple-frequency (BIS) analysis, 24h food recall, and blood samples were collected at 2-time points: diagnosis (T0) and after one month of completion of therapy (T1) for the main study group and one-time point for the control group. T-tests or Mann-Whitney Wilcoxon Test was used to compare variables. Linear regression analysis was conducted to test if PhA is related to the dependent variables after adjusting for age and body mass index.

### Results

119 women were included (61 with breast cancer and 58 healthy). There was no difference between the groups concerning anthropometrics, fat mass, and fat-free mass. Breast cancer patients had a worsening in PhA (p<0.001) after chemotherapy completion. PhA was positive statistically correlated with extracellular water, albumin, and the antioxidant markers at both times. The linear model showed that PhA was significantly predicted by C reactive protein, 2,2-Diphenyl-1-picrylhydrazyl (DPPH), Malondialdehyde (MDA), total body water/extracellular water, and body mass index fat mass. This model explained 58% of PhA variability (p<0.001).

### Conclusion

Our findings show that PhA is an easy and affordable tool that correlates oxidative stress markers in breast cancer patients, regardless of age or body mass index.

07963-0 and FAPESP scholarship number: 2019/
09877-9. The funders had no role in study design,
data collection and analysis, the decision to
publish, or the preparation of the manuscript.

**Competing interests:** The authors have declared
that no competing interests exist.

## Introduction

Breast cancer is the most common cancer in women (2.3 million new cases in 2021) and the
second among all populations worldwide [1]. Nearly 30% of all new diagnoses will progress
and become metastatic diseases [2]. Oxidative stress is a condition promoted by an imbalance
between oxidants and antioxidants, with increased reactive oxygen species (ROS) levels [3].
This condition has been shown to play an essential role in the pathogenesis of cancer, which
can be related to the development, proliferation, and progression of metastatic cancer cells [4].

Besides, ROS can be linked to inflammation through immune cell recruitment and cytokine
production that triggers inflammatory pathways, promoting chronic inflammation [5, 6].
Moreover, chronic inflammation is also key to developing several diseases such as diabetes,
cardiovascular disease, metabolic syndrome, neurodegeneration, ageing, cancer, and its pro-
gression [5–10]. In addition, oxidative stress can alter body composition by losing muscle
mass and strength, promoting sarcopenia [11, 12], a prognosis factor among cancer patients
[13, 14]. For breast cancer patients, it can be especially critical when sarcopenia is combined
with obesity (i.e., sarcopenic obesity), increasing mortality [13, 15].

Adding to the promotion of an oxidative environment, antineoplastic agents potentially
increase oxidative stress by the elevation of peroxidation and reduction of antioxidant nutri-
ents and enzymes [16, 17], which may explain the bad outcomes related to the after-treatment,
especially regarding metabolic alterations, eg: risk for cardiovascular disease, metabolic syn-
drome [18–22]. Previous research conducted by our group showed a deterioration in nutri-
tional status, physical function, visceral adiposity markers and development of metabolic
syndrome post-chemotherapy in early breast cancer patients [23, 24]. Thus, tracking oxidative
stress and inflammation markers are essential to identify individuals with greater risk for fur-
ther complications. However, there are several limitations regarding monitoring ROS, which
involve time-consuming, expensive, and complex techniques and the need for qualified staff
and facilities to perform laboratory analyses, limiting its use in clinical practice.

To overcome these limitations, phase angle (PhA), obtained from bioelectrical impedance
analysis (BIA), is a simple, fast, non-invasive, and affordable technique that has been explored
as a potential measurement to screen for oxidative stress and inflammation impairments [25–
30]. PhA is considered an indicator of cellular health [31], related to health issues such as mal-
nutrition [32, 33], loss of physical function [24], poor prognoses, and mortality [34–39], and
also reflects the hydration status [40], and is associated with the extracellular and intracellular
water ratio (ECW/ICW) [41].

In a narrative review conducted in 2021, the authors discussed the potential role of PhA as a
marker of oxidative stress, which might be justified due to its capacity to identify cellular integ-
rity and damage, which may occur as a ROS outcome [42]. Also, the cellular injuries promoted
by ROS might lead to water/fluid disbalance and a decrease in body cell mass, impacting cell
membrane conductivity and, thus, PhA results [42]. Although this result seems promising,
only a few studies have explored this field; therefore, there is a lack of evidence to confirm it.

Accordingly, the study aimed to examine the relationship between PhA and inflammatory and
oxidative stress biomarkers in women with breast cancer before and one month after chemother-
apy. We also further explored any possible alterations in the markers promoted by chemotherapy.

## Methods

### Study population

A prospective study was performed with women newly diagnosed with early stages breast can-
cer. Patients were recruited through clinical oncology practices at Mastology ambulatory of

General Hospital. This study received approval from the Mastology committee of the hospital and the Institutional Review Board at the University of Sao Paulo, General Hospital, approved the current study, obtained informed written consent, according to guidelines and standards for research involving human beings, regulated by Resolution 466/12 of the National Health Council (Protocol: HCRP 14608/2017).

During the clinical orientation of chemotherapy, a responsible nurse informed the patient about the study, those who were interested know more about it were forwarded to talk with the study researcher. All women who met the following inclusion criteria were enrolled in the study: age ≥18 years and <65 years; a histologically confirmed diagnosis of early breast cancer (range of stage I–III); and very first chemotherapy treatment course. Smokers, patients metabolic syndrome; worse blood pressure control, which means the use of two or more antihypertensive drugs; lipid disorders, which means values above the normal range for triglycerides, total cholesterol and low dense cholesterol, according to the criteria of the National Cholesterol Education Program's Adult Treatment Panel III (NCEP-ATP III) [43] and, the Brazilian nation recommendations [44]; diabetes type I or II or with a more recent glucose test above 125 mg/dl according to the results available on the electronic clinical records; pregnant women; who previously has already received or started chemotherapy in any other moment of life; those fitted with a defibrillator, cardiac pacemaker, metal implants or those with a local infection/wound preventing the use BIA pads, those unable to use a handheld dynamometer due to a neuromuscular disorder were all excluded.

Besides, once no cut-off for oxidative paraments has been proposed, we also include a control group of women with no history of cancer or chemotherapy treatment in this study. For this group, the same exclusion criteria were applied.

Women in control groups were recruited at the same hospital, the participants were employees or graduate student from the School of Medicine of Ribeirao Preto, Sao Paulo, Brazil. For this group potential participants were weighed and measured to determine BMI and completed a Health Status Screening Form to determine if they had any prior cancer or were under hormone or any other medication which could modify the metabolism that would have excluded them from participating in the study. The inclusion for this group were: female (sex-matched) > 18 years old <60 years old, no history of cancer and exclusion criteria were the same applied to the breast cancer group: smokers, patients with MetS, with worse blood pressure control, which means the use of two or more antihypertensive drugs, lipid disorders, which means values above the normal range for triglycerides, total cholesterol and low dense cholesterol according to the criteria of the National Cholesterol Education Program's Adult Treatment Panel III (NCEP-ATP III) [43] and, the Brazilian nation recommendations [44], with any type of diabetes (type I, type II, gestational) or with a more recent glucose test above 125 mg/dl according to the results available on the electronic clinical records, HIV, thyroid disease that is not currently managed with medication, pregnancy, BIA exclusion factor. After screening, the participants eligible for the study were scheduled for the data collection visit.

## Data collection

We collected data at baseline, before starting chemotherapy (T0), until one month after completing the treatment (T1), totalizing eight cycles of chemotherapy. The last evaluation was made 1 month after finalizing the chemotherapy and before the hormone therapy started, due to the possible association between hormone therapy and the increase of metabolic alterations [45]. We recruited participants from July 1st, 2017, to December 30th, 2018, and the data collection was completed in July 2019. Bioelectrical impedance multiple-frequency (BIS) and blood chemical analyzes were assessed at the T0 and T1. Also, food records were collected 4

different times. Socioeconomic, demographic, and therapeutic, were collected directly from patients using questionnaires or obtained from medical records. Written informed consent was obtained at the baseline visit. No patient in the breast cancer group had lymphedema.

A single evaluation was conducted for the control group, and the participants underwent the same assessment as the breast cancer participants group. A second food record was collected by a phone call a month later. For all methods used to assess the participants, was asked fasting for 12 hours previously.

### Anthropometric assessments

Anthropometric characteristics that were measured include body weight and body height, as proposed by Lohman [46]. Body mass index (BMI) was calculated as the ratio between the body weight and the height squared ($kg/m^2$). Interpretation of these results followed the international classification proposed by the World Health Organization [47]. The anthropometric measurements were recorded as the average of three consecutive measures.

### Bioelectrical impedance analysis

Body composition was assessed by using the bioelectrical impedance multiple-frequency (BIS) analysis (Body Composition Monitor–Fresenius Medical Care®), with different frequencies (5 to 1,000 kHz), and were considered the values obtained in 50 kHz. Patients were in supine decubitus, at rest 15 minutes before the measurement allowing a balance of body fluids with arms and legs abducted within a 30–45˚ angle from the trunk and electrodes to be affixed to the right hand and foot. BIS was calibrated regularly as an electronic verification module (usually provided by the manufacturer of this device). Other recommendations: refrain from any intense physical activity four hours prior to measurement, ensure that no metals are in the clothing, clean the skin with alcohol before placement of the electrodes.

The electrodes was positioned as follows: an opposite pair next to breast cancer surgery, being a distal electrode at the base of the middle finger and the proximal between the medial and lateral malleoli, away from 5 cm between them; The other pair was positioned in the contralateral hand the breast cancer surgery, with the distal electrode at the base of the middle finger and the proximal electrode coinciding with the style of style, also with a distance of 5 cm [48].

The BIS analysis provided data regarding resistence (R), reactance (Xc), fat mass (FM $_{BIS}$), lean tissue mass (LTM) which consists of sum of lean tissue excluding bone mineral content, phase angle (PhA), total body water (TBW), extracellular water (ECW) and intracellular water (IW). The accuracy of BIS has been previously demonstrated [49]. Additionally, fat-free mass (FFM $_{Equation}$) which consists of LTM plus bone mineral content and FM (FM $_{Equation}$) were obtained by predictive equation for white and non-white subjects proposed by Kotler et al. (1996) using DXA as the reference method [50].

**Women**: FFM = 0, 88 x [height $(cm)^{1.97}$ / impedance $(\Omega)^{0.49}$ x 1.0/22.22] + 0.081 + weight + 0.07

### Dietary data collection

Dietary data collection occurred using a 4-dietary recall of 24 hours for study, for the breast cancer group and 2 for the control group. The specific time frame was from the time the participant awoke in the morning to the time they slept at night. For this method was used the methodology of the triple pass 24-hour recall according with Nightingale et al. [51] to improve the accuracy for quantification of the recall. The results obtained by the recall was inserted in

the nutritional software Diet Box® to calculate the total of amount of energy and macronutrients ingested. This software uses the Brazilian table of food composition in the assessment.

Reported values were analyzed by the Multiple Source Method program (MSM) to estimate the usual intake distribution for daily consumed nutrients. The MSM is a statistically method proposed for use in Europe by a German team [52] and is accessible through an online platform open source in which by the probability of consumption and the amount consumed and regressions models correct the within-person variance of the food intake results obtained by the record and generate the usual intake for each participant [53]. Prior studies have shown that the MSM is a useful tool that provides usual nutrient and food intake estimates [53, 54], thus in order to improve the accuracy of the food consumption collected data, the MSM was applied.

## Blood biochemical analysis

Venous blood samples were collected for the blood biochemical analysis after fasting for 12 hours previously. A nurse during the hospital blood collection collected a 9ml tube of peripheral blood containing heparin in T0 and T1. The samples were processed in the nutrition and metabolism laboratory. The peripheral blood collected were centrifuged at 1000 g for 15min. After centrifugation, an aliquot of 200 μL was immediately acidified with 800 μL 5% trichloroacetic acid (TCA) for later vitamin C assay.

The remaining plasma and the plasma–trichloroacetic acid aliquot were stored in Eppendorf tubes at –80˚C for later analysis. Serum was used for the following analysis: Albumin (AL); C-reactive Protein (CRP); For the oxidative stress biomarkers evaluation, were analyzed: Malondialdehyde (MDA) for lipoperoxidation. For the antioxidant biomarkers evaluation, were analyzed: 2,2-diphenyl-1-picrylhydrazyl (DPPH); Glutathione (GSH); Seric tocopherol, retinol, and vitamin C.

AL was evaluated by Bromocresol Green albumin assay kit, MDA was determined by high-performance liquid chromatography (UV/VIS SPD-20A Shimadzu, Kyoto, Japan) [55] and it was used as lipid peroxidation marker. GSH was determined with the method developed by Rahman et al. [56] and CRP was determined by the latex immunoturbidimetric assay. Vitamin C was determined according to Roe & Kuether, 1943 [57] and tocopherol and retinol were performed by high-pressure liquid chromatography (HPLC).

All biochemical determinations were performed in duplicate and presented a mean variation of <5%.

## Statistical analysis

The sample size calculation was performed using G*Power software version 3.1.9.4, taking into consideration the effect of independent variables on PhA [58]. For a linear regression model, considering a large effect size of 0.35 showed that with a significance level of 95% and statistical power of 80%, the minimum number of participants required was 43. According to Cohen's guidelines, $f2 \geq 0.02$, $f2 \geq 0.15$, and $f2 \geq 0.35$ represent small, medium, and large effect sizes, respectively [59].

This study was a post hoc analysis of data from an ongoing study which aim to explore possible changes in body composition, metabolic and oxidative stress parameters [60, 61]. All women included in this study, consistently maintained all data collection appointments and, nobody left the study, therefore, we had no missing data within participants for static analysis.

Data are described as mean ± standard deviation. The normality (Shapiro-Wilk) and homogeneity of variances (Levene) of all variables were tested ($p > 0.05$). The variables were compared using T-tests or Mann-Whitney Wilcoxon Test depending on the distribution of the

data. The correlations of oxidative damage with the biochemical and BIS parameters and body compartments were evaluated using Pearson's or Spearman correlation coefficient depending on the distribution of the data. The strength of the correlation was classified as very weak for r < 0.19, weak for $0.20 \leq r < 0.39$, moderate for $0.4 \leq r < 0.59$, strong for $0.6 \leq r < 0.79$, and very strong for $r \geq 0.80$. Multiple regression analysis was conducted to further test whether PhA was predicted by the independent variables. To assess the ability of regressions models making predictions, it was used the verification by the least square methods. A p value < 0.05 was considered statistically significant for all tests. SAS studio was used for all statistical analyzes.

## Result

During the recruitment, we identified 180 new diagnostics for breast cancer patients. Fifty-seven were excluded due to diabetes, dyslipidemia, metabolic syndrome, or high blood pressure. Eighteen were excluded for metastatic breast cancer, and two were due to cognitive impairments. Four women did not want to join the study, and 14 were excluded due to absence at the scheduled collection visit, no fasting at the data collection visit, or starting chemotherapy before at baseline data collection schedule. The final sample included in this study was 61 women with early stages breast cancer, 6.6% stage I (n = 4), 59% stage II (n = 36) and 35% stage III (n = 21), which more than half were younger than 50 years (63.3%) and 65.6% premenopausal at recruitment (n = 40).

The prescribed protocol of treatment was the combination between Doxorubicin, Cyclophosphamide, and Docetaxel (AC-T) according to the Brazilian Society of Clinical Oncology guidance, which recommends the combination of 4 cycles of Doxorubicin 60 mg / m2 IV + cyclophosphamide 600 mg / m2 IV every 21 days, followed by four cycles of docetaxel 100 mg / m2 IV every 21 days [62, 63].

For the control group, 61 women were recruited. However, three did not attend the data collection visit. Therefore, the final sample included was 58 women. Control and breast cancer patients, both groups presented overweight according to BMI. We did not find statistically significant alteration in body weight, BMI, FM, or GSH levels during the follow-up period, but FFM (p < 0.001) and TBW (p = 0.01) were statistically significant. Chemotherapy also impacted PhA, EW, EW ratio, AL, CRP and HB (p<0.05). MDA and DPPH improved, and alpha-tocopherol increased after one month of chemotherapy treatment (p<0.05) (Table 1).

Regarding the comparison between breast cancer patients and the control group, there is no difference between age, weight, height, BMI, FM, FFM, LTM or TBW (Table 1). The control group presented healthier values for PhA, EX, EX ratio, AL, CRP, MDA, DPPH and GSH when compared to both times (T0 and T1), and all were statistically significant (p<0.05). The non-breast cancer participants also presented lower serum alpha-tocopherol and retinol values and better food ingestion, with lower calories, carbohydrates, total and saturated fat and higher protein, fiber, and vitamin E intake (p<0.05). Table 1 shows the complete data.

PhA had a statistically significant correlation for both times, T0 and T1, with variables related to body composition, nutritional status, and oxidative stress. PhA was statistically significantly correlated only with body composition parameters for the control group. In T0, PhA was positively correlated with LTM, TBW, IW, AL and GSH and negatively correlated to FM (p<0.05). For T1, the significant correlations were with EW, AL, HB and DPPH. For the control group, PhA was correlated to LTM, FM TBW and IW (p<0.05). Table 2 has a complete description of all correlations.

**Table 1. Sample characteristics and comparison among time and groups.**

| Variable | T0 | T1 | CG | p- value T0 x T1 | p-value (T0 x CG) | p-value T1 x CG |
|---|---|---|---|---|---|---|
| Age (years) | 46.50 (SD = 9.85) | - | 43.37 (SD = 9.74) | - | 0.08 | 0.08 |
| Weight (kg) | 71.70 (SD = 12.6) | 73.50 (SD = 12.6) | 76.35 (SD = 19.75) | 0.84 | 0.24 | 0.61 |
| BMI (kg/m2) | 28.54 (SD = 5.46) | 28.95 (SD = 4.37) | 28.57 (SD = 6.90) | 0.84 | 0.22 | 0.55 |
| LTM $_{BIS}$ (kg) | 34.00 (SD = 7.1) | 32.50 (SD = 5.6) | 35.62 (SD = 6.38) | 0.97 | 0.46 | 0.06 |
| FM $_{BIS}$ (kg) | 28.82 (SD = 9.09) | 28.78 (SD = 8.94) | 30.23 (SD = 13.82) | 0.84 | 0.51 | 0.64 |
| FFM $_{Equation}$ (kg) | 45.10 (SD = 5.1) | 47.50 (SD = 6.8) | 46.80 (SD = 5.6) | **<0.001** | 0.07 | 0.56 |
| FM $_{Equation}$ (kg) | 27.60 (SD = 10.3) | 27.30 (SD = 10.5) | 29.60 (SD = 15.6) | 0.62 | 0.32 | 0.27 |
| PhA | 6.05 (SD = 0.75) | 5.16 (SD = 0.77) | 6.35 (SD = 0.81) | **<0.001** | **0.03** | **<0.001** |
| TBW (L) | 31.90 (SD = 5.12) | 33.22 (SD = 6.22) | 33.37 (SD = 5.91) | **0.01** | 0.38 | 0.89 |
| ECW (L) | 14.35 (SD = 2.20) | 16.00 (SD = 3.09) | 14.77 (SD = 3.00) | **0.001** | 0.38 | **0.03** |
| TBW/ECW | 0.45 (SD = 0.02) | 0.48 (SD = 0.02) | 0.44 (SD = 0.02) | **<0.001** | 0.06 | **<0.001** |
| Energy (kcal) | 1775 (SD = 725) | 1700 (SD = 490) | 1240 (SD = 250) | 0.50 | **<0.001** | **<0.001** |
| Carb (g) | 235.48 (SD = 103.55) | 233.92 (SD = 81.12) | 67.59 (SD = 18.80) | 0.92 | **<0.001** | **<0.001** |
| Protein (g) | 80.41 (SD = 39.68) | 76.40 (SD = 23.14) | 147.25 (SD = 41.70) | 0.49 | **<0.001** | **<0.001** |
| Fat (g) | 56.81 (SD = 31.08) | 53.09 (SD = 19.30) | 42.13 (SD = 6.09) | 0.42 | **0.01** | **<0.001** |
| Sat fat (g) | 18.08 (SD = 12.05) | 17.41 (SD = 5.83) | 14.06 (SD = 13.45) | 0.37 | 0.44 | **<0.001** |
| Col (mg) | 264.87 (SD = 207.67) | 277.01 (SD = 129.27) | 251.71 (SD = 112.56) | 0.7 | 0.97 | 0.12 |
| Fiber (g) | 17.49 (SD = 11.17) | 15.35 (SD = 6.50) | 37.87 (SD = 14.31) | 0.19 | **<0.001** | **<0.001** |
| Vit A (mcg) | 405.11 (SD = 1115) | 424.41 (SD = 200.13) | 128.77 (SD = 58.12) | **<0.001** | 0.18 | **<0.001** |
| Vit E (mg) | 6.73 (SD = 5.25) | 8.31 (SD = 0.85) | 434.53 (SD = 397.78) | 0.0013 | **<0.001** | **<0.001** |
| Vit C (mg) | 150.46 (SD = 115.06) | 15.35 (SD = 7.05) | 11.06 (SD = 4.95) | **0.001** | **<0.001** | 0.75 |
| Sel (mcg) | 42.12 (SD = 37.55) | 44.35 (SD = 5.97) | 5.85 (SD = 2.82) | **0.03** | **<0.001** | **<0.001** |
| AL (g/dL) | 3.97 (SD = 0.65) | 3.44 (SD = 0.64) | 3.87 (SD = 0.46) | **<0.001** | 0.35 | **<0.001** |
| HB (g/dL) | 12.84 (SD = 1.30) | 11.40 (SD = 1.19) | - | **<0.001** | - | - |
| CRP (mg/dL) | 7.35 (SD = 13.73) | 15.94 (SD = 31.47) | 10.13 (SD = 8.97) | **0.05** | **0.02** | 0.17 |
| MDA | 7.89 (SD = 1.99) | 5.27 (SD = 2.66) | 4.05 (SD = 1.30) | **<0.001** | **<0.001** | **0.002** |
| GSH | 0.18 (SD = 0.04) | 0.20 (SD = 0.09) | 0.21 (SD = 0.05) | 0.42 | **0.03** | 0.30 |
| DPPH | 36.71 (SD = 17.02) | 45.08 (SD = 16.28) | 73.80 (SD = 16.13) | **0.01** | **<0.001** | **<0.001** |
| retinol (μM) | 1.55 (SD = 4.34) | 1.59 (SD = 3.93) | 1.40 (SD = 4.35) | 0.56 | **0.05** | **0.01** |
| alpha tocopherol (μM) | 22.46 (SD = 7.37) | 28.91 (SD = 9.24) | 17.82 (11.36) | **< .0001** | **0.01** | < .0001 |
| vitamin C (mg/dL) | 2.46 (SD = 2.02) | 1.35 (SD = 1.08) | 1.14 (SD = 1.47) | **0.05** | **< .0001** | **< .0001** |

T0: before starting chemotherapy, T1: until one month after completing the treatment, CG: control group, BMI: Body mass index, LTM: Lean tissue mass, FFM $_{Equation}$:
Fat-free mass obtained by the predictive equation [50], FM $_{Equation}$: Fat Mass obtained by the FFM predictive equation (51), PhA: Phase angle. TBW: Total body water.
ECW: Extracellular water. ECW/TBW: the ratio between extracellular water and total body water. Carb: Carbohydrate, Sat fat: Saturated fat, Vit A: Vitamin A, Vit E:
Vitamin E; Vit C: Vitamin C, AL: Albumin, CRP: C reactive protein, HB: Hemoglobin, MDA: Malondialdehyde, GSH: Glutathione, DPPH: α-diphenyl-β-picrylhydrazyl
* The mean difference is significant at a level of 0.05.

It was performed a multiple regression model to determine how much the PhA variation may be explained by body composition, nutritional, biochemical and stress oxidative parameters for both times. In T0, the model showed that AL (Beta = 0.004, p = 0.03), TBW/ECW (Beta = 0.16, p<0.001), BMI (Beta = 0.001, p = 0.0002), and FM (Beta = 0.0009, p = 0.00078) explained 49% of PhA variability (p<0.001). In T1, PhA was significantly predicted by CRP (Beta = 0.00005, p = 0.05), AL (Beta = 0.00302, p<0.001), MDA (Beta = -0.00111, p = 0.05), DPPH (Beta = 0.00022, p = 0.02), TBW/ECW (Beta = -0.19577, p<0.001), BMI (Beta = 0.00177, p<0.001), and FM (Beta = -0.00049, p = 0.00025) and this model explained 58% of PhA variability (p<0.001). The complete data are presented in Table 3.

**Table 2. Pearson correlation of phase angle and other studies variable.**

| Pears | T0 | | T1 | | CG | |
|---|---|---|---|---|---|---|
| | r | p | r | p | r | p |
| Weight | 0.02 | 0.82 | 0.23 | 0.06 | -0.10 | 0.42 |
| LTM $_{BIS}$ | 0.60 | **<0.001** | 0.21 | 0.10 | 0.67 | **<0.001** |
| FM $_{BIS}$ | -0.49 | **<0.001** | -0.12 | 0.35 | -0.51 | **<0.001** |
| TBW | 0.37 | 0.003 | -0.01 | 0.89 | 0.35 | 0.007 |
| EW | -0.01 | 0.90 | -0.25 | 0.05 | -0.12 | 0.36 |
| IW | 0.55 | **<0.001** | 0.19 | 0.14 | 0.61 | **<0.001** |
| AL | 0.29 | **0.02** | 0.25 | **0.05** | -0.02 | 0.37 |
| HB | 0.03 | 0.77 | 0.27 | **0.03** | - | - |
| CRP | 0.17 | 0.19 | 0.23 | 0.07 | 0.01 | 0.91 |
| MDA | -0.008 | 0.94 | -0.17 | 0.19 | 0.03 | 0.78 |
| DPPH | 0.01 | 0.91 | 0.3 | **0.01** | 0.04 | 0.75 |
| GSH | 0.25 | **0.05** | -0.02 | 0.83 | 0.04 | 0.73 |
| retinol | 0.44 | **0.0008** | 0.16 | 0.22 | 0.34 | 0.36 |
| alfa tocopherol | 0.17 | 0.19 | -0.14 | 0.29 | 0.10 | 0.67 |
| vitamin C | -0.11 | 0.40 | -0.27 | **0.03** | -0.04 | 0.85 |

LTM $_{BIS}$: Lean tissue mass provided by BIS analysis, FM $_{BIS}$: Fat Mass provided by BIS analysis, TBW: Total body water. ECW: Extracellular water. ICW: Intracellular water. AL: Albumin, CRP: C reactive protein, HB: Hemoglobin, MDA: Malondialdehyde, GSH: Glutathione, DPPH: α, α-diphenyl-β-picrylhydrazyl. Model adjusted for age and body mass index (BMI).

## Discussion

The main finding of the present investigation was the significant positive association between PhA and antioxidants agents (DPPH, retinol and GSH) after adjusting for age and BMI. In our model of linear regression analyses, the measures of body composition as FM and ECW/TBW, BMI and biochemical markers as AL, CRP, MDA and DPPH accounted for 49% of the variance in the PhA in T0 and 58% in T1. To our knowledge, only a few articles aimed to make

**Table 3. Multiple linear regression analysis of variables influencing the phase angle in T0 and T1.**

| Coefficients | T0 | | | T1 | | |
|---|---|---|---|---|---|---|
| | Beta | Standard error | P-value | Beta | Standard error | p-value |
| Intercept | 7.31012 | 0.34821 | < .0001 | 5.48285 | 0.35212 | < .0001 |
| CRP | * | * | * | 0.00010 | 0.00005 | **0.05** |
| ALB | 0.004971 | 0.002314 | **0.03** | 0.01331 | 0.00302 | < .0001 |
| MDA | * | * | * | -0.00111 | 0.00057 | **0.05** |
| DPPH | * | * | * | 0.00022 | 0.00010 | **0.02** |
| ECW/TBW | 0.168136 | 0.02563 | < .0001 | -0.19577 | 0.02548 | < .0001 |
| BMI | 0.001926 | 0.000492 | **0.0002** | 0.00177 | 0.00042 | < .0001 |
| FM $_{BIS}$ | 0.000904 | 0.000328 | **0.0078** | -0.00049 | 0.00025 | **0.05** |
| **Multiple R squared** | 0.49 | | | **Multiple R squared** | 0.58 | |
| **Adjusted R square** | 0.45 | | | **Adjusted R square** | 0.53 | |
| **P-value** | < .0001 | | | **P-value** | < .0001 | |

* FFM $_{BIS}$, CRP, MDA and DPPH were removed from the model in T0 by backward elimination selection. BMI: Body mass index, FM $_{BIS}$: Fat Mass provided by BIS analysis, TBW: Total body water. EW: Extracellular water. IW: Intracellular water. ECW/TBW: the ratio between extracellular water and total body water. AL: Albumin, CRP: C reactive protein, MDA: Malondialdehyde, GSH: Glutathione, DPPH: α-diphenyl-β-picrylhydrazyl. Model adjusted for age.

similar analyses. We are the first study exploring the relationship between PhA and oxidative stress parameters among breast cancer patients. Indeed, the PhA is a promising health parameter. Our review identified 16 studies that reported an association between PhA and direct and indirect inflammatory biomarkers [42]. Also, a cutoff to predict increased CRP levels has already been proposed [27].

Although the results for oxidative stress are still less expressive, our results agree with previous studies that evaluated this relationship with PhA. Recently, our research group identified that despite fewer studies have evaluated the relationship between PhA and markers of oxidative stress, available data suggest that PhA has potential to be used as an indicator (for screening) of oxidative damage [61]. Zouridakis et al. in 2016 reported a positive correlation between PhA and total antioxidant capacity (TAC) [27], and Venâncio et al. in 2021 found a negative correlation with advanced oxidation protein products [28]. In addition, another Brazilian group described a positive correlation between PhA and catalase, total radical-trapping antioxidant potential and a negative correlation with ferrous oxidation-xylenol orange (FOX) and AOPP [25, 26]. Our results only found the association between PhA and oxidant, antioxidant, CRP, and AL in the breast cancer group. For healthy populations, body composition parameters are the main determinants of PhA (Table 2). The same pattern is observed in the linear model, where after PhA deterioration in T1, post-chemotherapy, the biomarkers contributed to the model (Table 3).

According to Norman et al. (2012), in a healthy population, PhA is mainly determined by age, sex, and BMI [64], which concords with our results. The PhA concept is based on changes in resistance and reactance as alternating current passes through evaluated tissues. Therefore, the measured PhA depends on several biological factors such as the quantity of cells with their respective cell membranes, cell membrane integrity, and related permeability and the amounts of extracellular and intracellular fluids [41].

In the presence of diseases, additional parameters can impact PhA. Compared to a healthy population, PhA in disease states is usually lower and might be affected by infection, inflammation, or other disease-related parameters [64, 65]. Moreover, considering the body composition determinants on PhA, FFM, and extracellular and intracellular water might exert a more substantial effect [41]. It can be explained by the fact that PhA is a cellular integrity marker; consequently, cell membrane rupture can affect the equilibrium of water in the cell [66], which can elucidate the relation between PhA and ECW/ ICW.

Therefore, the alterations in the PhA associated with malnutrition, specially early phases of malnutrition that FFM loss has not still occurred, extracellular fluid expansion can lead to an increase in the ECW:ICW ratio leading to a decrease in the PhA. Our results also found a correlation between PhA and BIS's fluids components.

In this study, we did find differences between anthropometrics results and the main body composition variables (i.e., FFM and TBW). Furthermore, we still observed an important deterioration in healthy markers like ECW/TBW, PhA, AL and CRP. ECW/TBW ratio is considered a valuable tool to detect water variation. Thus, it is regarded as an index of edema, an expected adverse effect after chemotherapy treatment, which also can impact FFM values. Its change can be related to malnutrition and electrolyte irregularities and might be modified in an obesity scenario [66, 67]. In addition, water fluctuations impact PhA results [41] and, therefore, can be related to nutritional status; both parameters play an important role in cancer care [68]. After chemotherapy, PhA dropped to below the cutoff value associated with lower breast cancer survival ($\leq$5.6º) proposed by Gupta et al. [69]. Cornejo-Pareja et al. (2021) demonstrate that a cut-off PhA value less than 3.94 (ROC curve of survival) is more sensitive prognostic factor in predicting mortality in COVID-19 patients than standard biochemical measurements of inflammation such as ferritin, prealbumin, albumin, CRP. The authors highlight that this

results, on the general COVID-19 population and not limited only to critical patients, may have greater applicability in clinical practice. Additionally, the survival analysis revealed 2.48 times higher hazards ratio of mortality for a decrease in 1° in PhA value [70].

Regarding oxidative stress markers, it was observed an improvement in T1. The MDA levels, a product of lipidic oxidation, decreased at the same time that total oxidant capacity (DPPH) and glutathione (GSH) increased. Also, this alteration might be due to modification in serum alpha-tocopherol levels, which increased simultaneously. Alpha-tocopherol is a fat-soluble vitamin and can be considered one of the most potent antioxidants, which protect from ROS damage, especially the lipid peroxyl radicals [71, 72].

We hypothesized that the organism responded to the oxidant's growth, which was characterized by increased alpha-tocopherol levels. In this scenario, the liver mobilized its fat-soluble vitamin stores to regulate oxidative stress to a physiological level. These serum antioxidant changes are not observed for vitamin C, a water-soluble vitamin that is not stored in the body. Evidently, this oxidative liver regulation may change in a long-term response when the liver stores are consumed. Despite hepatic vitamin E mobilization to restore oxidative balance, the inflammatory process continues in this sample, evidenced by the higher levels of PCR and lower levels of AL.

The results of our study also indicate that, regardless of similarity in age, weight, BMI, and FM (all were not statistically significant), it was possible to verify differences in health markers. Breast cancer patients presented worse PhA, AL, food consumption and higher oxidative markers (i.e., fewer antioxidants and more oxidants species), which were more discrepant after chemotherapy treatment. These results are confirmed by other studies that have already reported an oxidative impairment among cancer patients compared to a control group [60, 73–75].

Despite lower oxidative stress, the control group had a higher level of CRP compared to T0 and no differences compared to T1. We believe the reason is the body composition profile of the control group. The control group presented higher body weight and FM, which might increase inflammation levels, especially for FM [76]. Although it is known there is a relationship between inflammation and oxidative stress [77], it did not promote higher ROS in the control group. It might be related to higher levels of oxidants (which this group had), a healthier diet, as shown in Table 2 and the level of physical activity, which unfortunately was not explored in this study.

Additionally, food intake was assessed only twice in the control group. For both groups, a 24h food recall was used, which may not capture the actual daily eating habits of participants. As a limitation, we also did not evaluate energy expenditure, which would allow us to understand better the differences observed in body composition and food intake between the groups.

Our interest in exploring alternative screening tools for oxidative stress is justified due to its involvement in the physiopathology of various diseases [6, 77]. In this context, potentially, PhA could be a tool that would be easily integrated into routine patient care as it is an affordable, non-invasive, simple method but effective in identifying those who would take advantage of a targeted behavioral approach.

## Strengths and limitations

The present study is not without limitations. The sample size was small and did not explore energy metabolism or physical activity level. We used a dietary recall to examine food intake, which is not a gold standard but applied the Multiple Source Method to increase the accuracy of the data. Still, the dietary records may not capture participants' actual daily eating habits, especially concerning micronutrients and under-reporting remains an important limitation of

self-reported dietary intake. A limitation of the design of this study is that the GC group was not followed for a similar time (only T0 period) as the test group, with the absence of comparable control data. Although no significant change is expected in the data of the control group, as this group did not receive any type of intervention, the absence of repeated measures in the control group does not allow for the assessment of time-related changes in parameters in the test group, regardless of treatment or disease status.

The strengths of this study include originality; only a few groups have studied this subject so far, the prospective approach, and the inclusion of a control group with strict inclusion criteria. Further studies are needed to investigate the association between PhA and oxidative stress and extrapolate these findings to other populations, ages, and sex. Additionally, there are no cutoff values for oxidative stress disorders, and a PhA's cutoff to screen oxidative stress has not been proposed yet. Finally, it is necessary to note that there is a lack of generalizability of these finds once there is a large variability in PhA values obtained from different BIA devices [78, 79].

## Conclusion

Our results suggest that breast cancer patients have worse nutritional status, food consumption, biochemical blood markers and oxidative stress biomarkers than a control group with similar age and body composition. Chemotherapy promoted a deterioration in PhA, increased inflammation by PCR and a higher mobilization of antioxidant regulatory mechanisms. PhA was statistically correlated to oxidative stress parameters regardless of age and BMI. Thus, PhA might be a potential inexpensive alternative to monitor oxidative stress in breast cancer patients. In-depth studies are needed to confirm these findings.

## Acknowledgments

We thank all of the research group on Nutrition and Breast Cancer of the University of São Paulo, especially the students who assisted in all phases of the study.

## Author Contributions

**Conceptualization:** Bruna R. da Silva, Mirele S. Mialich, Thais Gozzo, Alceu A. Jordão.

**Formal analysis:** Bruna R. da Silva, Loris P. Cruz.

**Funding acquisition:** Alceu A. Jordão.

**Investigation:** Bruna R. da Silva, Sarah Rufato, Loris P. Cruz.

**Methodology:** Bruna R. da Silva, Loris P. Cruz.

**Project administration:** Mirele S. Mialich, Alceu A. Jordão.

**Supervision:** Mirele S. Mialich, Thais Gozzo, Alceu A. Jordão.

**Writing – original draft:** Bruna R. da Silva, Sarah Rufato.

**Writing – review & editing:** Mirele S. Mialich, Thais Gozzo, Alceu A. Jordão.

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
