## [Decision Letter · Decision Letter 0]

22 Aug 2022

PONE-D-22-16707Phase angle is related to oxidative stress and antioxidant biomarkers in breast cancer patients undergoing chemotherapy.PLOS ONE

Dear Dr. Ramos da Silva,

Thank you for submitting your manuscript to PLOS ONE. After careful consideration, we feel that it has merit but does not fully meet PLOS ONE’s publication criteria as it currently stands. Therefore, we invite you to submit a revised version of the manuscript that addresses the points raised during the review process.

The manuscript is interesting, but both reviewers pointed out important aspects to be adjusted. Note the considerations pointed out in the methods and statistical treatment and in the results and conclusions. I also recommend proofreading in English throughout the text.

We look forward to receiving your revised manuscript.

Kind regards,

Alvaro Reischak-Oliveira, Ph.D.

Academic Editor

PLOS ONE

“Bruna Ramos da Silva was founded by São Paulo Research Foundation (FAPESP). Grant number: 2017/07963-0 and FAPESP fellowship Grant number: 2019/09877-9”

Reviewers' comments:

Reviewer's Responses to Questions

**Comments to the Author**

1. Is the manuscript technically sound, and do the data support the conclusions?

Reviewer #1: Yes

Reviewer #2: Partly

2. Has the statistical analysis been performed appropriately and rigorously? 

Reviewer #1: No

Reviewer #2: No

3. Have the authors made all data underlying the findings in their manuscript fully available?

Reviewer #1: Yes

Reviewer #2: Yes

4. Is the manuscript presented in an intelligible fashion and written in standard English?

Reviewer #1: Yes

Reviewer #2: Yes

5. Review Comments to the Author

Reviewer #1: The authors aimed to examine the association between PhA and inflammatory and oxidative stress biomarkers in women with breast cancer before and after one month of chemotherapy. The authors should be congratulated by the relevance of the research. However, several methodological issues should be addressed:

1. Please clarify if inclusion criteria involves active smoking status, metabolic syndrome, hypertension. 

2. More details about the recruitment and screening process of the control group are required (ex, where were they recruited from? Other medical appoitments? How was the physical level of CG females? Did this women also present the metabolic syndrome and its features and were active smokers?

3. Before describing the methods used to assess the participants, clarify if a fasted condition was assured.

4. In the BIA description more details are required about the protocol (position, electrodes, etc). 

5. Also, provide information about the raw BIA parameters that were assessed (resistance, reactance, PhA) and which frequency was used.

6. Please include the precision (coefficient of variation-CV, for instances) for body composition variables and specifically for the PhA. The authors mentioned the precision for the biochemical variables but did not state which measurement was used (was it the CV?) and was it the same across all variables?

7. Please include a reference that supports the expected effect size of 0.35.

8. If the authors are interested in exploring how PhA could be a simple marker of health-related parameters (in this case oxidative stress and inflammation), the multiple regression analysis should consider PhA as the independent variable and not the dependent variable. The authors want to test the potential usefulness of PhA in tracking health-related parameters after adjusting for confounding factors (age, weight and height), because it is a simple marker, and not the other way around. Please make the required changes in the results and discussion.

Minor comment: please correct the word "statically" throughout the manuscript.

Reviewer #2: The manuscript is generally well written although in places the English is not entirely clear. Some examples are noted below. A thorough reading and sub-editing would be beneficial. Points to be considered.

Intro, para 1, line 5 "pathogenesis"

Intro, para 2, line 2 "that triggers inflammatory"

Intro, [ara 3, line 3 "the bad outcomes related to the aftertreatment, especially regarding metabolic alterations ". This is not clearly worded - what is actually meant by "after-treatment"? An example of a metabolic alteration would be useful.

Methods, population, para 3, last line Please provide some details about the control group recruitment. Were the age, sex-matched etc.?

Data collection, para 1. Please provide some information in relation to time of diagnosis, surgical treatment etc., presence of lymphedema (known to effect impedance measurements)....

Anthropometric measurements. To what resolution were these measurements made, in replicate...? More detail required.

BIA. BIA is an indirect assessment method that is known to be population specific and requires careful standardization of the measurement procedures (see doi: 10.1080/03091902.2017.1333165. this is in children but the same principles apply in all age groups). Please provide details. The device used is a BIS device, hence a PhA is available at all measurement frequencies. Make it clear that this study is using PhA at 50 kHz (I assume!).

Statistical analysis, last para. "t-test" not "T-test". It is not entirely clear but it seems that the test group were assessed twice (To and T1) but the control group once only. In this case, the test group differences should be assessed using a paired-tests and the CG versus test group using a group analysis. A weakness of this study design is that the CG group were not followed for a similar time period as the test group. Thud change in the test group may simply be a reflection of normal fluctuation but unknown in the absence of comparable control data. This needs to be addressed and discussed as study limitation.

Multiple regression analysis was used but what type, was it simple OLS or a derivative e.g. ridge regression or even a Bayesian approach. More details are required. For example which of the many potential variables were included and why? How was multi-collinearity assessed?

Statistical analysis, penultimate sentence. "To assess the ability of regressions models making predictions, it was used the verification by the least square methods". What exactly does this mean? Assessment of predictive power requires some form of cross-validation, e.g., split-grroup, LOOCV, K-means etc. This appears not to have been undertaken. Table 3 indicates that a particular set of variables may have some value in terms of explaining variability but this is not the same of developing a predictive model or algorithm. Please be more explicit in what you are wishing to achieve.

Results, Table 1. Please present all data to a consistent number of significant places, i.e., one. Also the the column headings, e.g., "T0xT1" requires more explanation. Are the values "t " values?, "P" values? Be consistent with units, e.g., both kg and KG are used. Use SI units. Abbreviations are not consistent, e.g., ECW and EX.

The BIA data do not seem to be consistent. For example, mean FFM at T0 = 34 kg for CG = 35.6 kg, The corresponding TBW values are 31. 9 and 33.4 L. BIA is a 2C model and is typically calibrated against deuterium dilution to provide TBW and then FFM calculate using the hydration fraction of FFM (0.732). THus the calculated FFM from TBW are 43.6 and 45.6 for CG. These are not the FFM values provided? Why the discrepancy? THis casts a cloud over the quality of the data. This must be explained.

Discussion. The authors discuss the relationship between PhA and various variables in their study and those of others. They are, however, not duly critical that these are simply empirically-observed relationships that may have tenuous biological or physiological links. Although, not specifically stated I assume that the data are whole body (wrist to ankle) impedance data. Thus the PhA value represents some form of average value arising from all cells and structures along the conductive path. This will be all variety of tissue types including, muscles, organs etc. Consequently, it is difficult to see a real (causal?) connection between PhA and, for example, circulating albumin. This point needs to be made. Study limitations are not discussed, particularly, the absence of repeat measurements in the control group to allow for time-related changes in parameters in the test group irrespective of treatment or disease status.

6. PLOS authors have the option to publish the peer review history of their article (what does this mean?). If published, this will include your full peer review and any attached files.

Reviewer #1: No

Reviewer #2: No

---

## [Author Response · Author response to Decision Letter 0]

24 Oct 2022

Thank you very much for the opportunity to review our manuscript “Phase angle is related to oxidative stress and antioxidant biomarkers in breast cancer patients undergoing chemotherapy”. We believe that all reviewer suggestions have substantially contributed to a better presentation of our findings. We thank them for their time and expertise.

---

## [Decision Letter · Decision Letter 1]

29 Nov 2022

PONE-D-22-16707R1Phase angle is related to oxidative stress and antioxidant biomarkers in breast cancer patients undergoing chemotherapy.PLOS ONE

Dear Dr. Ramos da Silva,

Thank you for submitting your manuscript to PLOS ONE. After careful consideration, we feel that it has merit but does not fully meet PLOS ONE’s publication criteria as it currently stands. Therefore, we invite you to submit a revised version of the manuscript that addresses the points raised during the review process. ==============================

ACADEMIC EDITOR: Reviewer 2 raised some intersting points that must be considered. Please submit your revised manuscript by 16th December. If you will need more time than this to complete your revisions, please reply to this message or contact the journal office at plosone@plos.org. Please include the following items when submitting your revised manuscript:A rebuttal letter that responds to each point raised by the academic editor and reviewer(s). You should upload this letter as a separate file labeled 'Response to Reviewers'.A marked-up copy of your manuscript that highlights changes made to the original version. You should upload this as a separate file labeled 'Revised Manuscript with Track Changes'.An unmarked version of your revised paper without tracked changes. You should upload this as a separate file labeled 'Manuscript'.If applicable, we recommend that you deposit your laboratory protocols in protocols.io to enhance the reproducibility of your results. Protocols.io assigns your protocol its own identifier (DOI) so that it can be cited independently in the future. For instructions see: https://journals.plos.org/plosone/s/submission-guidelines#loc-laboratory-protocols. Additionally, PLOS ONE offers an option for publishing peer-reviewed Lab Protocol articles, which describe protocols hosted on protocols.io. Read more information on sharing protocols at https://plos.org/protocols?utm_medium=editorial-email&utm_source=authorletters&utm_campaign=protocols.

We look forward to receiving your revised manuscript.

Kind regards,

Alvaro Reischak-Oliveira, Ph.D.

Academic Editor

PLOS ONE

Journal Requirements:

Reviewers' comments:

Reviewer's Responses to Questions

**Comments to the Author**

1. If the authors have adequately addressed your comments raised in a previous round of review and you feel that this manuscript is now acceptable for publication, you may indicate that here to bypass the “Comments to the Author” section, enter your conflict of interest statement in the “Confidential to Editor” section, and submit your "Accept" recommendation.

Reviewer #1: All comments have been addressed

Reviewer #2: (No Response)

2. Is the manuscript technically sound, and do the data support the conclusions?

Reviewer #1: Yes

Reviewer #2: Partly

3. Has the statistical analysis been performed appropriately and rigorously? 

Reviewer #1: Yes

Reviewer #2: No

4. Have the authors made all data underlying the findings in their manuscript fully available?

Reviewer #1: Yes

Reviewer #2: Yes

5. Is the manuscript presented in an intelligible fashion and written in standard English?

Reviewer #1: Yes

Reviewer #2: Yes

6. Review Comments to the Author

Reviewer #1: The authors addressed the raised comments. The paper was improved and is now able to provide additional insights to the related research field

Reviewer #2: The authors have addressed most of the issues raised but I remain concerned about the BIA data in Table 1, i.e., the discrepancy between the stated TBW and FFM values. The authors have provided a brief but comprehensive review of the BIA methods and made reference to differences in hydration fraction in different populations. But this does not address the key issue. Going back to my example from the Table, for a TBW of 31.9 L and an FFM of 34 kg suggests an HF of 31.9/34 or 93.8%, physiologically unrealistic. Hydration values such as this are simply not seen.

Something is wrong here. I may be misinterpreting the data but I do not think so. To state that the papers focus is not on FM and FFM is missing the point. If these are not correct then how can the reader be assured that the phase data are correct since FFM and FM have been calculated from the measured BIA data. This MUST be explained for, at least, this reader to have confidence in data validity.

7. PLOS authors have the option to publish the peer review history of their article (what does this mean?). If published, this will include your full peer review and any attached files.

Reviewer #1: No

Reviewer #2: No

---

## [Author Response · Author response to Decision Letter 1]

19 Dec 2022

Thank you very much for this more opportunity to review our manuscript “Phase angle is related to oxidative stress and antioxidant biomarkers in breast cancer patients undergoing chemotherapy”. 

Please find below a point-by-point response to reviewer’s comments. 

JOURNAL REQUIREMENTS:

Reviewer #1: The authors addressed the raised comments. The paper was improved and is now able to provide additional insights to the related research field.

Thank you again for this opportunity. We believe that all reviewer suggestions have substantially contributed to a better presentation of our findings.

Reviewer #2: The authors have addressed most of the issues raised but I remain concerned about the BIA data in Table 1, i.e., the discrepancy between the stated TBW and FFM values. The authors have provided a brief but comprehensive review of the BIA methods and made reference to differences in hydration fraction in different populations. But this does not address the key issue. Going back to my example from the Table, for a TBW of 31.9 L and an FFM of 34 kg suggests an HF of 31.9/34 or 93.8%, physiologically unrealistic. Hydration values such as this are simply not seen.

Something is wrong here. I may be misinterpreting the data but I do not think so. To state that the papers focus is not on FM and FFM is missing the point. If these are not correct then how can the reader be assured that the phase data are correct since FFM and FM have been calculated from the measured BIA data. This MUST be explained for, at least, this reader to have confidence in data validity.

Thank you for these important appointments on the BIA data (Table 1). We are really honored by this review and the opportunity to study and reflect further on our study’s data.

BIA is a doubly indirect method which uses predictive equations to estimate body composition derived from comparisons with reference methods. Most studies using BIA to predict body composition in the last years developed many equations. 

Talking specifically of the device used in this study, we used the FM and FFM obtained directly from the BIA. We checked the manual and consulted the Fresenius website to confirm which equation the software uses; however, we could not verify that.

In this sense, our research group entirely agrees with the reviewer's comments. Instead of using the FM and FFM obtained from the device, we correct it by using the BIA's raw data and the predictive equation developed by Kotler et al. (1996).

Kotler et al. (1996) proposed a new BIA formula validated against DXA for white and non-white subjects, including a group of HIV-positive patients, which used logarithmic transformation of height, reactance, and impedance and found them to be more accurate predictors than equations using height2/resistance.

Going back to the relevant reviewer's comments, BIA is a 2C model and is typically calibrated against deuterium dilution to provide TBW and then FFM calculate using the hydration fraction of FFM (0.732). The corresponding TBW values are 31.9 L, 33,2 L and 33.4 L, respectively in T0, T1 and CG.

Therefore, FFM and FM have been calculated from predictive equation (Kotler et al., 1996) and the final values obtained are perfectly compatible with the values expected by the BIA and correspond to the acceptable hydration fraction (HF) according to the principles of this method.

 Kotler et al. (1996) BIS

 FFM Equation (kg) FM Equation (kg) HF FFM BIS (kg) FM BIS (kg)

T0 45,1 (SD 5,1) 27,6 (SD 10,3) 0,705 (SD 0,05) 34 (SD 7,1) 28,82 (SD 9,09)

T1 47,5 (SD 6,8) 27,3 (SD 10,5) 0,697 (SD 0,08) 32,5 (SD 5,6) 28,78 (SD 8,94)

CG 46,8 (SD 5,6) 29,6 (15,6) 0,711 (SD 0,07) 35,62 (SD 6,3) 30,23 (SD 13,82)

All these new FFM and FM values were incorporated in the manuscript, as well as their respective updated p values.

[1] Kotler DP, Burastero S, Wang J, Pierson RN. Prediction of body cell mass, fatfree mass, and total body water with bioelectrical impedance analysis: effects of race, sex, and disease. Am J Clin Nutr 1996;64(suppl). 489S-97S.

---

## [Editor Report · Decision Letter 2]

20 Dec 2022

PONE-D-22-16707R2Phase angle is related to oxidative stress and antioxidant biomarkers in breast cancer patients undergoing chemotherapy.PLOS ONE

Dear Dr. Silva,

Thank you for submitting your manuscript to PLOS ONE. After careful consideration, we feel that it has merit but does not fully meet PLOS ONE’s publication criteria as it currently stands. Therefore, we invite you to submit a revised version of the manuscript that addresses the points raised during the review process.

Although one of the reviewers has already accepted the manuscript, the second reviewer still highlights an aspect that deserves attention.

He remains concerned about the BIA data in Table 1, i.e., the discrepancy between the stated TBW and FFM values. His concern makes sense, and so I'd like you to respond carefully to him.

Please submit your revised manuscript by Feb 04 2023 11:59PM. If you will need more time than this to complete your revisions, please reply to this message or contact the journal office at plosone@plos.org. Please include the following items when submitting your revised manuscript:A rebuttal letter that responds to each point raised by the academic editor and reviewer(s). You should upload this letter as a separate file labeled 'Response to Reviewers'.A marked-up copy of your manuscript that highlights changes made to the original version. You should upload this as a separate file labeled 'Revised Manuscript with Track Changes'.An unmarked version of your revised paper without tracked changes. You should upload this as a separate file labeled 'Manuscript'.If applicable, we recommend that you deposit your laboratory protocols in protocols.io to enhance the reproducibility of your results. Protocols.io assigns your protocol its own identifier (DOI) so that it can be cited independently in the future. For instructions see: https://journals.plos.org/plosone/s/submission-guidelines#loc-laboratory-protocols. Additionally, PLOS ONE offers an option for publishing peer-reviewed Lab Protocol articles, which describe protocols hosted on protocols.io. Read more information on sharing protocols at https://plos.org/protocols?utm_medium=editorial-email&utm_source=authorletters&utm_campaign=protocols.

We look forward to receiving your revised manuscript.

Kind regards,

Alvaro Reischak-Oliveira, Ph.D.

Academic Editor

PLOS ONE
---

## [Author Response · Author response to Decision Letter 2]

26 Jan 2023

We completely agree with the review’s concerns. And due to the discrepancy between the TBW and FFM, we tried to contact the BIA’s manufacturer to check in every possible way which equation was built into the BIA’s system to estimate FM and FFM.

Unfortunately, this information is not disclosed in the manual. For this reason, we calculated FM and FFM from BIA’s raw data and the Kotler equation (1996), as described in detail in " Response to Reviewers" attached.

JOURNAL REQUIREMENTS:

Reviewer #1: The authors addressed the raised comments. The paper was improved and is now able to provide additional insights to the related research field.

Thank you again for this opportunity. We believe that all reviewer suggestions have substantially contributed to a better presentation of our findings.

Reviewer #2: The authors have addressed most of the issues raised but I remain concerned about the BIA data in Table 1, i.e., the discrepancy between the stated TBW and FFM values. The authors have provided a brief but comprehensive review of the BIA methods and made reference to differences in hydration fraction in different populations. But this does not address the key issue. Going back to my example from the Table, for a TBW of 31.9 L and an FFM of 34 kg suggests an HF of 31.9/34 or 93.8%, physiologically unrealistic. Hydration values such as this are simply not seen.

Something is wrong here. I may be misinterpreting the data but I do not think so. To state that the papers focus is not on FM and FFM is missing the point. If these are not correct then how can the reader be assured that the phase data are correct since FFM and FM have been calculated from the measured BIA data. This MUST be explained for, at least, this reader to have confidence in data validity.

Thank you for these important appointments on the BIA data (Table 1). 

BIA is a doubly indirect method which uses predictive equations to estimate body composition derived from comparisons with reference methods. Most studies using BIA to predict body composition in the last years developed many equations. 

Talking specifically of the device used in this study, we used the FM and FFM obtained directly from the BIA. We checked the manual and consulted the Fresenius website to confirm which equation the software uses; however, we could not verify that.

In this sense, our research group entirely agrees with the reviewer's comments. Instead of using the FM and FFM obtained from the device, we correct it by using the BIA's raw data and the predictive equation developed by Kotler et al. (1996).

Kotler et al. (1996) proposed a new BIA formula validated against DXA for white and non-white subjects, including a group of HIV-positive patients, which used logarithmic transformation of height, reactance, and impedance and found them to be more accurate predictors than equations using height2/resistance.

Going back to the relevant reviewer's comments, BIA is a 2C model and is typically calibrated against deuterium dilution to provide TBW and then FFM calculate using the hydration fraction of FFM (0.732). The corresponding TBW values are 31.9 L, 33,2 L and 33.4 L, respectively in T0, T1 and CG.

Therefore, FFM and FM have been calculated from predictive equation (Kotler et al., 1996) and the final values obtained are perfectly compatible with the values expected by the BIA and correspond to the acceptable hydration fraction (HF) according to the principles of this method.

All these new FFM and FM values were incorporated in the manuscript, as well as their respective updated p values - Manuscript (January 26) and Revised Manuscript with tracked changes (January 26).

We hope this effort will clarify these important points. We are really honored by this review and the opportunity to study and reflect further on our study’s data.

[1] Kotler DP, Burastero S, Wang J, Pierson RN. Prediction of body cell mass, fatfree mass, and total body water with bioelectrical impedance analysis: effects of race, sex, and disease. Am J Clin Nutr 1996;64(suppl). 489S-97S.

---

## [Decision Letter · Decision Letter 3]

17 Feb 2023

PONE-D-22-16707R3Phase angle is related to oxidative stress and antioxidant biomarkers in breast cancer patients undergoing chemotherapy.PLOS ONE

Dear Dr. Silva,

Thank you for submitting your manuscript to PLOS ONE. After careful consideration, we feel that it has merit but does not fully meet PLOS ONE’s publication criteria as it currently stands. Therefore, we invite you to submit a revised version of the manuscript that addresses the points raised during the review process.

I fully understand that the delay in accepting the manuscript can cause some frustration. However, reviewer 2 makes a point that needs to be clarified. The issue of the incorrect BIS TBW and FFM remains, so it is essential that you provide a proper explanation for this aspect.

Please submit your revised manuscript by Apr 03 2023 11:59PM If you will need more time than this to complete your revisions, please reply to this message or contact the journal office at plosone@plos.org. Please include the following items when submitting your revised manuscript:A rebuttal letter that responds to each point raised by the academic editor and reviewer(s). You should upload this letter as a separate file labeled 'Response to Reviewers'.A marked-up copy of your manuscript that highlights changes made to the original version. You should upload this as a separate file labeled 'Revised Manuscript with Track Changes'.An unmarked version of your revised paper without tracked changes. You should upload this as a separate file labeled 'Manuscript'.If applicable, we recommend that you deposit your laboratory protocols in protocols.io to enhance the reproducibility of your results. Protocols.io assigns your protocol its own identifier (DOI) so that it can be cited independently in the future. For instructions see: https://journals.plos.org/plosone/s/submission-guidelines#loc-laboratory-protocols. Additionally, PLOS ONE offers an option for publishing peer-reviewed Lab Protocol articles, which describe protocols hosted on protocols.io. Read more information on sharing protocols at https://plos.org/protocols?utm_medium=editorial-email&utm_source=authorletters&utm_campaign=protocols.

We look forward to receiving your revised manuscript.

Kind regards,

Alvaro Reischak-Oliveira, Ph.D.

Academic Editor

PLOS ONE

Journal Requirements:

Reviewers' comments:

Reviewer's Responses to Questions

**Comments to the Author**

1. If the authors have adequately addressed your comments raised in a previous round of review and you feel that this manuscript is now acceptable for publication, you may indicate that here to bypass the “Comments to the Author” section, enter your conflict of interest statement in the “Confidential to Editor” section, and submit your "Accept" recommendation.

Reviewer #2: All comments have been addressed

2. Is the manuscript technically sound, and do the data support the conclusions?

Reviewer #2: Yes

3. Has the statistical analysis been performed appropriately and rigorously? 

Reviewer #2: Yes

4. Have the authors made all data underlying the findings in their manuscript fully available?

Reviewer #2: Yes

5. Is the manuscript presented in an intelligible fashion and written in standard English?

Reviewer #2: Yes

6. Review Comments to the Author

Reviewer #2: Thank you for now calculating FFM according to the Kotler equation.

I do not wish to delay the paper further but the issue of the incorrect BIS TBW and FFM remains. The authors are correct about BIA being a 2C model etc., thus it is disturbing that BIS_FM + BIS_FFM do not equal weight (Table 1: 34 + 28.8 = 62.8 yet weight is 71.7 kg). I wonder if this is a nomenclature issue and that BIS-FFM is actually BIS_Lean, i.e. FFM - BMC. This could be the explanation. I suggest that the authors consult the device manual to check this.

This discrepancy needs to be explained and/or discussed. I am sorry about this but the paper must be scientifically correct.

7. PLOS authors have the option to publish the peer review history of their article (what does this mean?). If published, this will include your full peer review and any attached files.

Reviewer #2: No

---

## [Author Response · Author response to Decision Letter 3]

20 Feb 2023

We would like to thank you, the reviewer, for the opportunity to review our data further. Based on the reviewer’s comment, we have reviewed the Fresenius Medical Care Body Composition Monitor manual and website; indeed, there was a nomenclature issue. Fresenius Body Composition Monitor describes that the device provides information regarding Lean Tissue Mass and not Fat-free mass [1]. The device estimates the Lean Tissue Mass from Extracellular Water and Total Body Water information [1]. 

Thank you for bringing this to our attention. Throughout the manuscript, we have corrected this by replacing Fat-free Mass (FFM) obtained by the BIS device with the appropriate nomenclature: Lean Tissue Mass (LTM).

The information regarding Lean Tissue Mass can be found and confirmed at:

https://www.freseniusmedicalcare.com/en/body-composition-monitor

https://www.freseniusmedicalcare.com/fileadmin/data/masterContent/pdf/Healthcare_Professionals/Fluid_Management/BCM_Technical_Data.pdf

References

[1] Fresenius Medical Care. BCM - Body Composition Monitor. https://www.freseniusmedicalcare.com/en/body-composition-monitor (accessed February 16, 2023).

---

## [Decision Letter · Decision Letter 4]

6 Mar 2023

Phase angle is related to oxidative stress and antioxidant biomarkers in breast cancer patients undergoing chemotherapy.

PONE-D-22-16707R4

Dear Dr. Silva,

We’re pleased to inform you that your manuscript has been judged scientifically suitable for publication and will be formally accepted for publication once it meets all outstanding technical requirements.

Kind regards,

Alvaro Reischak-Oliveira, Ph.D.

Academic Editor

PLOS ONE

Reviewers' comments:

Reviewer's Responses to Questions

**Comments to the Author**

1. If the authors have adequately addressed your comments raised in a previous round of review and you feel that this manuscript is now acceptable for publication, you may indicate that here to bypass the “Comments to the Author” section, enter your conflict of interest statement in the “Confidential to Editor” section, and submit your "Accept" recommendation.

Reviewer #2: All comments have been addressed

2. Is the manuscript technically sound, and do the data support the conclusions?

Reviewer #2: Yes

3. Has the statistical analysis been performed appropriately and rigorously? 

Reviewer #2: Yes

4. Have the authors made all data underlying the findings in their manuscript fully available?

Reviewer #2: Yes

5. Is the manuscript presented in an intelligible fashion and written in standard English?

Reviewer #2: Yes

6. Review Comments to the Author

Reviewer #2: Thanks for checking and making the correction regarding Lean versus FFM. You are not the first who have been caught by this nomenclature error!

7. PLOS authors have the option to publish the peer review history of their article (what does this mean?). If published, this will include your full peer review and any attached files.

Reviewer #2: No

---

## [Editor Report · Acceptance letter]

8 Mar 2023

PONE-D-22-16707R4 

Phase angle is related to oxidative stress and antioxidant biomarkers in breast cancer patients undergoing chemotherapy. 

Dear Dr. da Silva:

I'm pleased to inform you that your manuscript has been deemed suitable for publication in PLOS ONE. Congratulations! Your manuscript is now with our production department. 

Kind regards, 

on behalf of

Dr. Alvaro Reischak-Oliveira 

Academic Editor

PLOS ONE